# Roof Fractures of Near-Vertical and Extremely Thick Coal Seams in Horizontally Grouped Top-Coal Drawing Method Based on the Theory of a Thin Plate

**Guojun Zhang** [1,2,3,*] **, Quansheng Li** [1] **, Zhuhe Xu** [1] **and Yong Zhang** [2,3,*]

1 State Key Laboratory of Water Resource Protection and Utilization in Coal Mining, National Institute of Clean and Low Carbon Energy, Beijing 102209, China

2 School of Energy and Mining Engineering, China University of Mining and Technology, Beijing 100083, China

3 Beijing Key Laboratory for Precise Mining of Intergrown Energy and Resources, China University of Mining and Technology, Beijing 100083, China

* Correspondence: 20039430@ceic.com (G.Z.); johnzy68@hotmail.com (Y.Z.)

**Abstract:** During the mining process of the near-vertical seam, there will be movement and collapse of the "roof side" rock layer and the "overlying coal seam," as well as the emergence of the "floor side" rock layer roof which is more complicated than the inclined and gently inclined coal seams, which causes problems with slippage or overturning damage. With the increase of the inclination of the coal seam, the impact of the destruction of the immediate roof on the stope and roadway gradually becomes prominent, while the impact of the destruction of the basic roof on the stope and the roadway gradually weakens. The destruction of the immediate roof of the near-vertical coal seam will cause a large area of coal and rock mass to suddenly rush to the working face and the two lanes, resulting in rapid deformation of the roadway, overturning of equipment, overturning of personnel, and even severe rock pressure disaster accidents, all of which pose a serious threat to coal mine safety and production. It is necessary to carry out research on the mechanical response mechanism of the immediate roof of near-upright coal seams, to analyse the weighting process of steeply inclined thick coal seam sub-level mining. A four fixed support plate model and top three clamped edges simply supported plate model for roof stress distribution are established before the first weighting of the roof during the upper and lower level mining process. The bottom three clamped edges simply supported plate model and two adjacent edges clamped on the edge of a simply supported plate model are established for roof stress distribution before periodic weighting of the roof during the upper and lower level mining process. The Galerkin method is used to make an approximate solution of deflection equation under the effect of sheet normal stress, and then roof failure criterion is established based on the maximum tensile stress strength criterion and generalized Hooke law. This paper utilizes FLAC$^{3D}$ finite element numerical simulation software, considering the characteristics of steeply inclined thick coal seam sub-level mining. It undertakes orthogonal numerical simulation experiment in three levels with different depths, coal seam angles, lateral pressure coefficient, and orientation of maximum horizontal principal stress, and translates roof stress of corresponding 9 simulation experiment into steeply inclined roof normal stress. We conclude that the distribution law of normal stress along dip and dip direction of a roof under the circumstance of different advancing distances and different sub-levels. The caving pace of first weight and periodical weight were counted under the effect of the roof uniform normal stress. It can better predict the weighting situation of the working face and ensure the safe, efficient, and sustainable mining of coal mines.

**Keywords:** near-vertical coal seams; horizontally grouped top-coal drawing method; fracture modes; coordinate conversion; the maximum tensile stress; the first fracture span; periodic fractures span





## 1. Introduction

With the gradual depletion of coal resources in mines in Eastern China and the contradiction between the coal resource exploitation and environment in the mines in Central China, the focus of coal resource exploitation was changed from Eastern China to Western China. Western China has a larger proportion of near-vertical (the seam inclination is greater than 45° in China) and extremely thick coal seams (the thickness of coal seams is greater than 8 m in China) [1,2], and the proportion continues to increase year by year. The coal seams with an inclination angle of 85° to 90° are near-vertical coal seams, which are widely found in Xinjiang, Anhui, Gansu, Inner Mongolia, Asturias in Spain, and the Lorraine coal areas in France [3]. The amplitude is bent, twisted, upright, or inverted, forming a nearly upright coal seam occurrence. The near-vertical coal seams are formed after many tectonic movements and evolutions, and their geological conditions and stress distributions are complex. The mining of a near-vertical coal seams forms a special coal rock structure, which is significantly different from other inclined coal seams.

The research on the mechanisms of deformation and fracture of a roof is divided into two approaches. In one approach, the goaf roof is simplified as an elastic beam, and the theory of the elastic beam is used to study the deformation and fracture mechanisms [4]. Qian [5] established voussoir beam theory based on the overlying strata and discussed the sliding-rotation stability condition of the voussoir beam structure. Song [6] analysed the basic top movement and breaking law of the horizontal sub-section top coal caving working face by using the rock slab theory and concluded that the pressure on the bottom side of the working face is lower than the pressure on the roof side and has a hysteresis phenomenon. Shi [7] analysed the difference of the coal caving process between the steeply inclined coal seam and the gently inclined coal seam and carried out an experimental study on the release performance of the broken top coal in the horizontal segmented top coal mining of the steeply inclined and extra-thick coal seam, appropriately increasing the segment height. Wang [8] studied the horizontal section height of steeply inclined fully mechanized caving face and listed the problems and solutions after reasonably raising the horizontal height. Cheng [9] used numerical calculation and similar simulation methods to compare the rock movement laws in the process of top coal caving mining with inclined layers and horizontal sections and believed that inclined layered mining could reduce the collapse of surrounding rock and the intensity of surface movement. Lai [10] obtained the range of pre-blasting and support by monitoring the loosening range of the top coal and comparing the stress-strain law of the top coal in the advanced pre-blasting. Dai [11] studied the rock movement mechanism of steeply inclined horizontal staged mining and established a prediction model of surface movement. It is believed that with the mining face being arranged layer by layer from top to bottom, the surface mobile basin has the value of expansion and subsidence to the roof side, as well as the characteristics of continuous accumulation. Cui [12] analysed the continuous disturbance effect of the sub-section caused by the horizontal section mining, and it is believed that increasing the thickness of the sub-section is conducive to weakening the impact on the lower sub-section. Shabanimashcool [13] presented two analytical approaches for studying voussoir beams by considering the horizontal loading condition of the beams. He [14] studied the elastic foundation coefficient, the span, and the stiffness of the main roof effect on the first fracture of the main roof with elastic foundation boundary. Guo [15] carried out effective support resistance and roof support technology of a fully mechanized mining face with hard roof conditions and thick coal seams. Yang [16] carried out evolution characteristics of overburden caving and void during multi-horizontal sectional mining in steeply inclined coal seams. He [17] carried out mechanism and prevention of rock burst in steeply inclined and extremely thick coal seams for fully mechanized top-coal caving mining. Su [18] conducted an experimental study on the rockburst and plate-pressing process of granite using a true triaxial test system and obtained the acoustic emission (AE) precursor characteristics of the instability of coarse-grained hard rock. Dong [19,20] confirmed that the reduction of isotropic components and the increase of double even

numbers can serve as precursors for rock fracturing development and proposed that the anisotropic characteristics of wave velocity changes and AE event rates are useful supplements for identifying rock fracturing. Chen [21] carried out collapse behaviour and control of hard roofs in steeply inclined coal seams. Kong [22] carried out a stability analysis of a coal face based on coal face-support-roof system in a steeply inclined coal seam. He [23] carried out a true triaxial test system to conduct experiments on the rockburst and plate-pressing process of granite and obtained the mechanisms and precursors of slip and fracture of coal-rock parting-coal structure.

Although many experts and scholars have done a great deal of research on the mining technology and equipment of steeply inclined and extra-thick coal seams, and have achieved a series of results, there are few studies on the roof problems faced in the process of mining of steeply inclined and extra-thick coal seams. Mining steeply inclined and extra-thick coal seams safely and efficiently have become urgent problems to be solved in the development of the Chinese coal industry. Many scholars generally believe that steeply inclined and extra-thick coal seams have the following characteristics in the process of mining. Firstly, during the mining process of the steeply inclined coal seam working face, with the increase of the coal seam inclination angle, the sliding force of the surrounding rock along the tangential direction (coal seam inclination direction) increases, and the normal vertical stress decreases. Secondly, when the working face inclination angle exceeds the natural repose angle of a caving rock accumulation, the caving broken rock will slide down and roll along the floor of the working face, thus forming a migration law different from the near-horizontal and gently inclined coal seam. Thirdly, along the working face inclination, there is a large difference in the filling degree (the lower part is filled and compacted, the middle part is filled, the upper part is not filled), and the gangue migration law. Fourthly, along the inclined direction, the bearing pressure distribution shows asymmetrical features with less stress because the pressure along the middle and upper part of the working face is large, while the lower part of the working face is not filled. After the steeply inclined coal seam working face is advanced for a certain distance, the floor rock strata within a certain range will intensify and move to the mined space, resulting in an increase in the trend of deformation and damage of the floor rock mass. If it is not restrained when the floor is damaged, slips may occur and this slip damage area develops upward along the inclined direction. This causes large-scale instability of the floor rock mass, resulting in a higher likelihood of the destruction of the support system composed of the "roof-support-floor" of the working face.

At present, the research on the equipment, technology, and surrounding rock control of the horizontally grouped top-coal drawing method (HGTC) of steeply inclined and extra-thick coal seams in China is not mature enough. The steeply inclined and extra-thick coal seam has the characteristics of an area with a large amount of coal mined. The stress concentrates around the stope and the higher degree and the mine pressure appears severe. With the continuous increase of coal mining depth, the problems of mine pressure and floor are more prominent, mainly in the following aspects: the side pressure of the roadway is large and the maintenance of the mining roadway is difficult. In particular, the roof and floor are unstable in large areas, and a large area of coal and rock mass can suddenly rush to the working face and two roadways, which will cause rapid deformation of the roadway, overturning of equipment, overturning of personnel, and even serious ground pressure disaster accidents. This is can make coal mines unsafe because high-efficiency production brings serious threats, therefore, it is very necessary to study the causes, forms and depths of roof damage in steeply inclined coal seams.

## 2. Diagrams of the Logical Relation

In this work, the roof fracture regularity of the near-vertical and extremely thick coal seams with HGTC was studied using the theory of an elastic thin plate. Four mechanical models of the immediate roof were established based on the different mining conditions before the roof fractures with the immediate roof assumed to be in a state of elasticity.

The orthogonal experiment designs were used to analyse the normal stress distribution of the immediate roof. The immediate roof normal stress distributions of the FLAC$^{3D}$ are substituted in the four thin plate models. By introducing the fracture criteria, the four models can be applied to simulate the fracture process of the immediate roof with HGTC. The diagram of the logical relation of the article analysis process is shown in Figure 1.

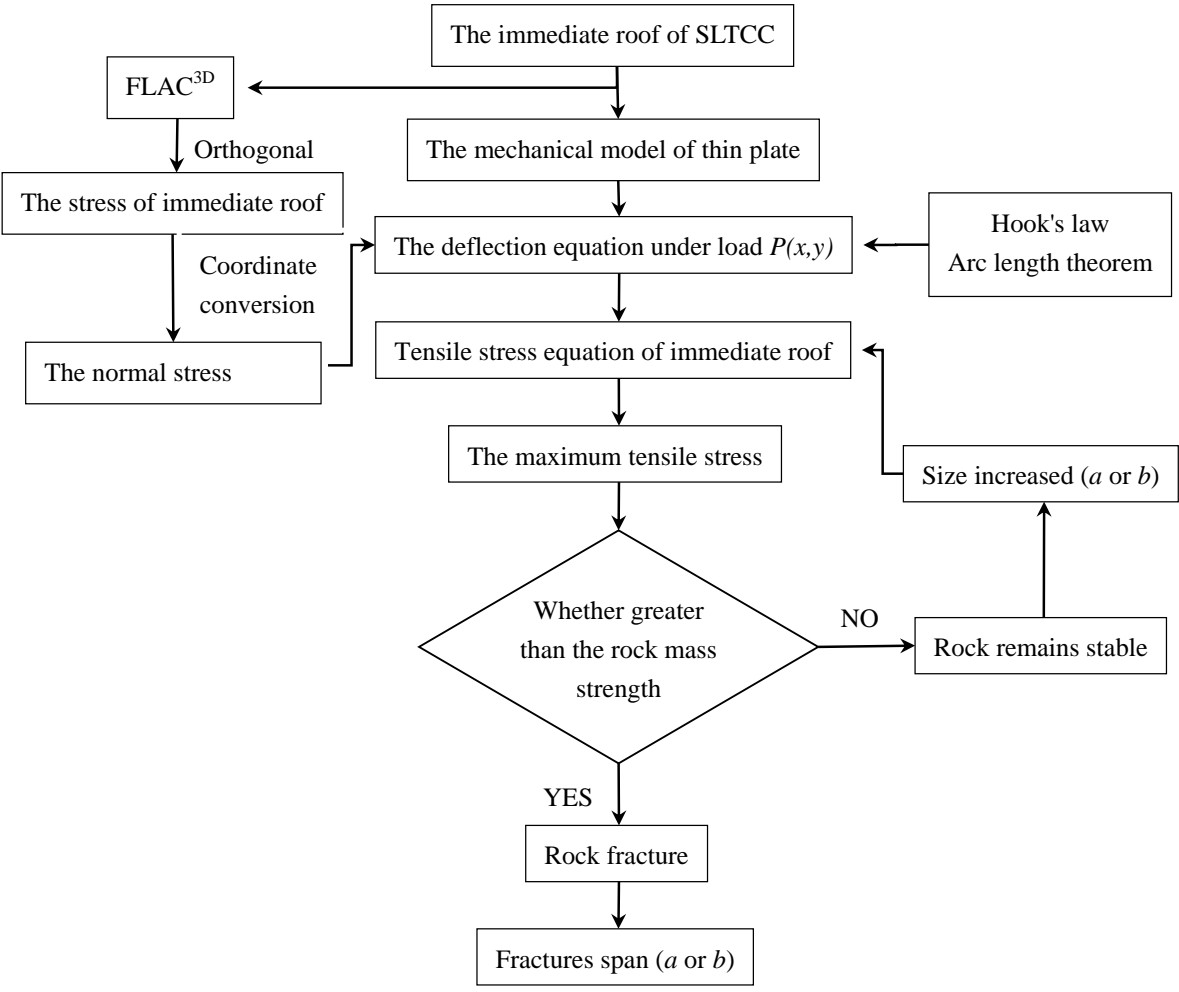

**Figure 1.** The diagrams of the logical relation.

## 3. Engineering Background

This work takes the Adaohai Mine as the research engineering background. Adaohai Mine is in the southern margin of the Daqingshan Coalfield of the Yinshan Mountains, 15 km away from Salazi Station and 7 km away from Beijing-Tibet highway in Inner Mongolia in China. The topography of the Adaohai Mine is very complicated, with steep cliffs, high mountains, deep valleys, and V-shaped gully development. The bedrock is exposed, the vegetation is very rare, and the coverage rate is low. The coal seams in the mining area are nearly vertical, and the ground collapse area formed in the goaf is small, only a narrow strip. However, with the gradual deepening of the mining depth, the range of ground collapse expands. More than 70% of the mining area is dominated by conglomerate and clastic rocks, with frequent phase changes, alternating fluvial facies and peat swamp facies, large changes in coal seam thickness, complex structure, and difficult to compare coal layers. The coal measure strata in the mining area include two coal seam groups, CU2 and CU4. The average dip angle of coal seams is 75°, and the maximum is 86°. The average thickness of the coal seam is 26 m. The height of subsection is 16 m, shown in Figure 2.

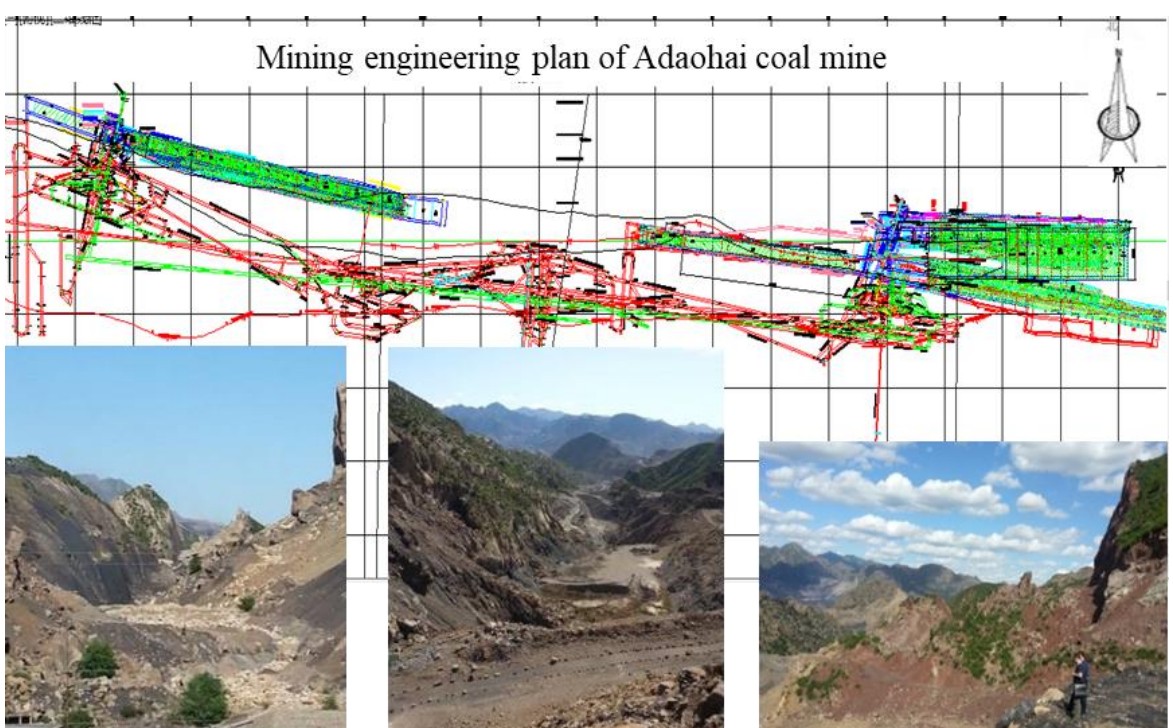

**Figure 2.** Mining pattern and surface subsidence of Adaohai coal mine in China.

The coal mining method is horizontally grouped top-coal drawing method mining, which developed from the flat slicing mining method. The difference between the two is that the height of each group for the former is several times that of each slice for the latter. The coal cutting method (mechanical cutting or blasting), similar to the method used in flat slicing, is accepted at the bottom of each group of coal, while the top coal caves by the aid of gravity. For a compact coal seam, which is hard to cave, a vibratory blast is used before drawing the top coal. This mining method possesses apparent advantages, like high monthly output for each working face and low divagate of roadways and so on, as shown in Figure 3.

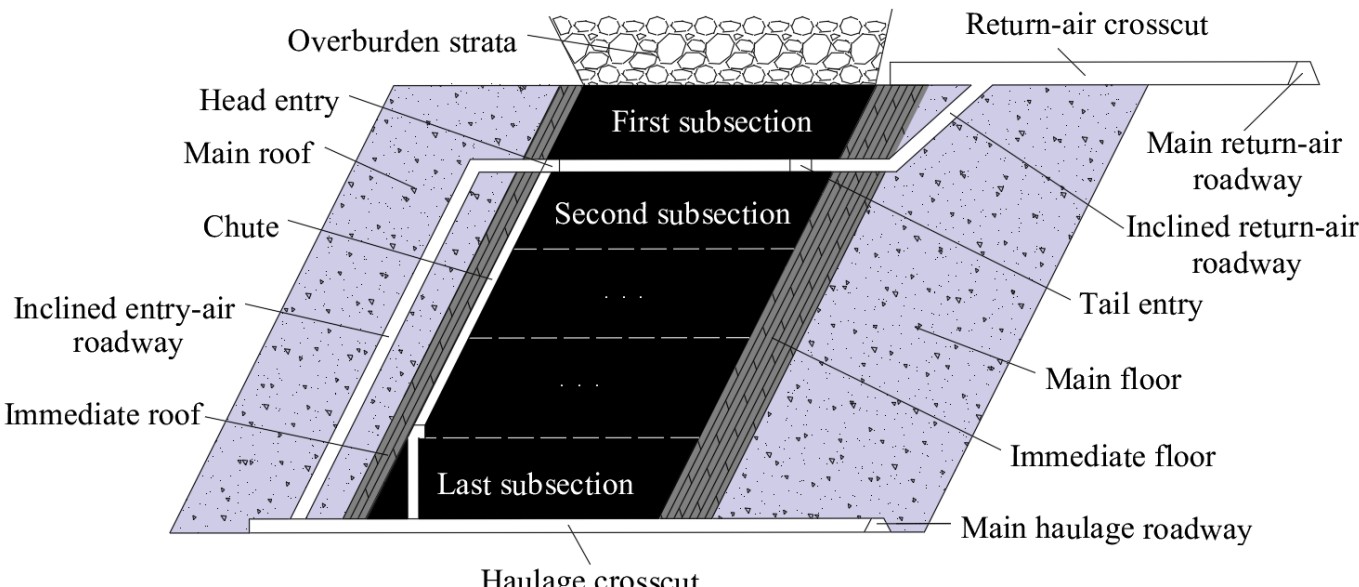

**Figure 3.** The diagrammatic sketch of HGTC.

The lithology and thickness of the roof and the bottom are shown in Table 1.

**Table 1.** The conditions of the roof and the bottom.

| Roof and Bottom | Lithology | Thickness |
|---|---|---|
| Main roof | Conglomerate | 25 m |
| Immediate roof | Sandstone | 2.0 m |
| Floor | Kaolin | 4.4 m |
| Basic bottom | Pebbly sandstone, Conglomerate | 28 m |

The ratio of the immediate roof thickness to the horizontal subsection height is 0.125. This ratio, less than 0.2 and more than 0.01, satisfies the assumptions of the thin plate theory [24]. Therefore, the thin plate mechanics models are introduced to analyse the stress distribution of the roof in various HGTC mining conditions.

## 4. Mechanical Model for the Roof of the HGTC Mining Face

### 4.1. Pressure Characteristics of the Near-Vertical Coal Seam Mining Process

The top of the inclined thick seam working face is a composite roof which is composed of rock stratums and coal seams. Because the strength of the coal seam is lower than the strength of the two sides of the rock, the destruction, caving, and release of the roof coal seam is preceded by roof rock, a large space is formed above the goaf. If the ratio of the goaf filling is low, the rock activity above the working face is more intense. Because the coal seam dip angle is large, the forces in the normal direction of the rock stratum (the interaction forces between rock stratums) have great variation after coal seam mining. The normal stress of the immediate roof adjacent to the exploitation space has an obvious change. The stress in inclination direction and trend direction of rock stratum varies small, and the rock layer above the top coal can form a relatively stable "articulated rock plate" and "pressure arch" structure. The space of the goaf will gradually enlarge with the enlargement of mining space, the "articulated rock plate" and "pressure arch" structure will gradually change from a steady state to an unstable state and then will be destroyed. The main fractures that form are bending break, rotation wreck, overall instability slipping, and horizontal movement and so on. The change of stress state in the immediate roof will directly influence the failure characteristics of the immediate roof.

The destruction of the rock layer is a combination of various forms of destruction in the working face mining of the inclined thick seam. The rock layers near the mining space (immediate roof and immediate floor) are more complicated, and the rock layers far away from the mining space (main roof and main floor) are relatively simple. The reason is that the stress field around the rock has become increasingly small with an increasing distance from the mining space. Therefore, the damage of the immediate roof will directly influence strata behaviour of the stope and laneway. Meanwhile, the working face pressure of the inclined thick seam is more complex and more intense than other conditions of occurrence.

### 4.2. The Thin Plate Mechanics Model in Different Mining Conditions

When the first subsection was excavated, the immediate roof experienced the first fracture and periodic fractures in sequence [25]. The four-edges clamped plate model (model A) was applied to calculate the stress distribution before the first fracture phase, as shown in Figure 4a. The three-edges clamped and forth-simple support plate model (model B) was applied to calculate the stress distribution during the periodic phase, as shown in Figure 4b.

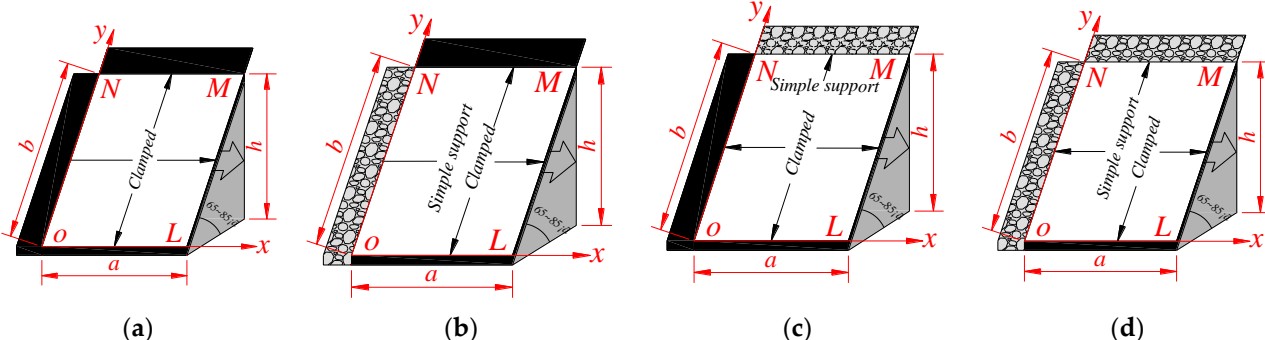

**Figure 4.** The mechanical model of the roof under different mining conditions (h is the subsection height, a is mining length, and b is the width of immediate roof). (**a**) model A; (**b**) model B; (**c**) model C; (**d**) model D.

When the first subsection was completed, mining began in the other subsections. The upper edge of roof, close to the goaf of upper subsection, was simplified as simple support. The immediate roof also experienced the first fracture and periodic fractures in turn. The first model is the bottom-three-clamped edges simply supported plate model (model C), and it was applied to calculate the stress distribution before the first fracture phase, as shown in Figure 4c. The other model has two adjacent edges clamped on the edge of a simply supported plate model (model D), and it was applied to calculate the stress distribution during the periodic phase, as shown in Figure 4d.

Model A and model C boundary conditions correspond to the first subsection and other subsection immediate roof first fracture event, respectively, and the first fracture pace of the immediate roof is equal to the length of "a". Model B and model D boundary conditions correspond to the first subsection and other subsection of the immediate roof, respectively, between immediate roof periodic fractures. The periodic fracture pace of the immediate roof is equal to the length of "a".

*4.3. Disturbance Equation of the Mechanics Models*

4.3.1. Galerkin Method

The bending problem of elastic thin plates can be solved directly and indirectly only in a few cases. The Galerkin method is another energy method based on an extreme value principle alongside a trigonometric series method [26].

The strain energy density of three-dimensional linear elastomer is [27]:

$$W = \frac{1}{2}(\sigma_{11}\varepsilon_{11} + \sigma_{22}\varepsilon_{22} + \sigma_{33}\varepsilon_{33} + \sigma_{12}\gamma_{12} + \sigma_{23}\gamma_{23} + \sigma_{31}\gamma_{31}) \tag{1}$$

where $W$ is the strain energy density of linear elastomer, J; $\sigma_{11}$, $\sigma_{22}$, $\sigma_{33}$, $\sigma_{12}$, $\sigma_{22}$, $\sigma_{31}$ is stress, Pa; $\varepsilon_{11}$, $\varepsilon_{22}$, $\varepsilon_{33}$ is volume strain; $\gamma_{12}$, $\gamma_{23}$, $\gamma_{31}$ is torsional strain.

For the small deflection bending of thin plates, according to the basic assumptions of thin plate theory $\varepsilon_{33}$, $\varepsilon_{32}$, $\varepsilon_{31}$, so its strain energy can be expressed as [27]

$$U = \iiint_v W\mathrm{d}V = \frac{1}{2}\iiint_v (\sigma_{11}\varepsilon_{11} + \sigma_{22}\varepsilon_{22})\mathrm{d}x_1\mathrm{d}x_2\mathrm{d}x_3 \tag{2}$$

where $U$ is the strain energy of linear elastomer, J.

The relationship between the stress and strain of the thin plate and the deflection of the plate can be obtained from the elasticity and Hooke's Law [27]:

$$\left.\begin{array}{l} \varepsilon_{11} = -x_3 \frac{\partial^2 w}{\partial x_1^2}, \sigma_{11} = -\frac{E}{1-v^2} x_3 \left( \frac{\partial^2 w}{\partial x_1^2} + v \frac{\partial^2 w}{\partial x_2^2} \right) \\[2mm] \varepsilon_{22} = -x_3 \frac{\partial^2 w}{\partial x_2^2}, \sigma_{22} = -\frac{E}{1-v^2} x_3 \left( \frac{\partial^2 w}{\partial x_2^2} + v \frac{\partial^2 w}{\partial x_1^2} \right) \\[2mm] \gamma_{12} = -x_3 \frac{\partial^2 w}{\partial x_1 \partial x_2}, \sigma_{22} = -\frac{E}{1+v} x_3 \frac{\partial^2 w}{\partial x_1 \partial x_2} \end{array}\right\} \tag{3}$$

where, $w$ is the deflection of elastic thin plate; $v$ is Poisson's ratio; $E$ is the elastic modulus, $N/m^2$.

Substituting Equation (3) into Equation (2), the strain energy expressed by deflection w after finishing is:

$$U = \frac{E}{2(1-v^2)} \iiint_v x_3^2 \left( \frac{\partial^2 w}{\partial x_1^2} + \frac{\partial^2 w}{\partial x_2^2} \right)^2 - 2(1-v) \left[ \frac{\partial^2 w}{\partial x_1^2} \frac{\partial^2 w}{\partial x_2^2} - \left( \frac{\partial^2 w}{\partial x_1 \partial x_2} \right)^2 \right] dx_1 dx_2 dx_3 \tag{4}$$

Integrate Equation (4) along the direction of plate thickness h to obtain

$$U = \frac{D}{2} \iint_F x_3^2 \left( \frac{\partial^2 w}{\partial x_1^2} + \frac{\partial^2 w}{\partial x_2^2} \right)^2 dx_1 dx_2 - D(1-v) \iint_F \left[ \frac{\partial^2 w}{\partial x_1^2} \frac{\partial^2 w}{\partial x_2^2} - \left( \frac{\partial^2 w}{\partial x_1 \partial x_2} \right)^2 \right] dx_1 dx_2 \tag{5}$$

where, $F$ is the length and width range of the elastic thin plate; $D$ is the bending stiffness of the elastic thin plate, $D = Eh^3/12(1-v^2)$, $N/m$; $h$ is elastic thin plates thickness, $m$.

For a rectangular plate with a fixed boundary, when $w = 0$ at the surrounding boundary, the second term integral on the right of Equation (5) is 0, so the strain energy of the rectangular plate with a surrounding boundary deflection of $w = 0$ is

$$U = \frac{D}{2} \iint_F x_3^2 \left( \frac{\partial^2 w}{\partial x_1^2} + \frac{\partial^2 w}{\partial x_2^2} \right)^2 dx_1 dx_2 \tag{6}$$

If the plate is only subjected to the normal load $P(x,y)$, the external force potential energy is

$$V = -\iint_F P(x,y) w dx_1 dx_2 \tag{7}$$

where, $V$ is the potential energy under the action of external force, J; $P(x,y)$ is the load distribution function of the elastic plate, Pa.

The combination of Equations (6) and (7) can obtain that the elastic potential energy of the whole plate is

$$\Pi = U + V = \frac{D}{2} \iint_F x_3^2 \left( \frac{\partial^2 w}{\partial x_1^2} + \frac{\partial^2 w}{\partial x_2^2} \right)^2 dx_1 dx_2 - \iint_F p(x,y) w dx_1 dx_2 \tag{8}$$

where $\Pi$ Is the elastic potential energy of elastic thin plate, J.

Potential energy of Equation (8) $\Pi$ the extreme value $w$ is the solution of the thin plate bending problem. When the boundary condition of the thin plate is complicated, it is difficult to directly solve the partial differential equation of the plate. The Galerkin method uses the energy method to solve the partial differential equation of the thin plate. In principle, the Galerkin method is the equivalent of applying the method of variation of parameters to a function space by converting the equation to a weak formulation. Typically, one then applies some constraints on the function space to characterize the space with a finite set of basic functions [26]. The Galerkin method provides a powerful numerical

solution for differential equations. To calculate the bending thin plate with the Galerkin method, the deflection of thin plate can be written as

$$\omega = \sum_{m=1}^{\infty} \sum_{n=1}^{\infty} A_{mn} \varphi_{ij} \tag{9}$$

where $\varphi_{ij}$ is the displacement function that meets all the displacement boundary conditions and stress boundary conditions. $A_{mn}$ indicates undetermined constants that satisfy the following:

$$\iint_{F} \left[ D\nabla^4 \omega - p(x_1, x_2) \right] \varphi_{ij} dx_1 dx_2 = 0 \tag{10}$$

The deflection equation of the four mechanical models is expressed in the space Cartesian coordinate system (SCCS), in which the *X*-axis is along the strike direction of the immediate roof, the *Y*-axis is along the inclined direction of the immediate roof, and the *Z*-axis is along the normal direction of the immediate roof.

### 4.3.2. Disturbance Equation of Model A

Model A is shown in Figure 4a using the SCCS. The *OL*, *LM*, *ON*, and *MN* are clamped by substance coal and rock. The boundary conditions of model A can be expressed as

$$\begin{cases} x = 0, x = a, \omega = 0, \frac{\partial \omega}{\partial x} = 0 \\ y = 0, y = b, \omega = 0, \frac{\partial \omega}{\partial y} = 0 \end{cases} \tag{11}$$

where $\omega$ is the deflection of thin plate, and $a$ and $b$ can be seen as constant.

According to the theory of elasticity [27,28] and the theory of plates and shells [29,30], the deflection of the thin plate of model A should satisfy

$$\frac{\partial^4 \omega}{\partial x^4} + 2\frac{\partial^4 \omega}{\partial x^2 \partial x^2} + \frac{\partial^4 \omega}{\partial y^4} = \frac{p(x,y)}{D} \tag{12}$$

where $p(x, y)$ is the normal stress loading on the thin plate, N.

According to the Galerkin method and the boundary conditions of the four-edges clamped plate, the deflection equation can be given by

$$\omega_1 = \sum_{m=1}^{\infty} \sum_{n=1}^{\infty} A_1 \left( 1 - \cos \frac{2m\pi x}{a} \right) \left( 1 - \cos \frac{2m\pi y}{b} \right), \ (m, n \in \mathrm{N}^+) \tag{13}$$

where $A_1$ is the undetermined constant of the model A, and $m$ and $n$ are positive integers.

Deriving Equation (12), substituting it into Equation (13), and then substituting the result into Equation (10), the undetermined constants of the model A can be obtained as

$$A_1 = \frac{a^3 b^3}{4\pi^4 D} \cdot \frac{\int_0^a \int_0^b p(x,y) \left( 1 - \cos \frac{2\pi x}{a} \right) \left( 1 - \cos \frac{2\pi y}{b} \right) dx dy}{3a^4 + 2a^2 b^2 + 3b^4} \tag{14}$$

### 4.3.3. Disturbance Equation of Model B

Model B is shown in Figure 4b using the SCCS. The *OL*, *LM*, and *MN* are clamped by substance coal and rock, and the *ON* is simple support. The boundary conditions of model B can be expressed as

$$\begin{cases} x = 0, \omega = 0, \frac{\partial^2 \omega}{\partial x^2} = 0 \\ x = a, \omega = 0, \frac{\partial \omega}{\partial x} = 0 \\ y = 0, y = b, \omega = 0, \frac{\partial \omega}{\partial y} = 0 \end{cases} \tag{15}$$

According to the Galerkin method and the boundary conditions of model B, the deflection equation can be given by

$$\omega_2 = \sum_{m=1}^{\infty} \sum_{n=1}^{\infty} A_2 \left(1 - \cos\frac{2n\pi y}{b}\right) \sin^3\frac{m\pi x}{a} \tag{16}$$

where $A_2$ is the undetermined constant of model B.

Deriving Equation (12) and substituting it into Equation (16), and then substituting the result into Equation (10), the undetermined constants of model B can be given by

$$A_2 = \frac{32a^3b^3}{\pi^4 Dm^3n^3} \cdot \frac{\int_0^a \int_0^b p(x,y)\left(1 - \cos\frac{2n\pi y}{b}\right)\sin^3\frac{m\pi x}{a}dxdy}{135(a/m)^4 + 72(a/m)^2(b/n)^2 + 80(b/n)^4} \tag{17}$$

### 4.3.4. Disturbance Equation of Model C

Model C is shown in Figure 4c using the SCCS. The *ON*, *LM*, and *MN* are clamped by substance coal and rock, and the *OL* is simple support. The boundary conditions of model A can be expressed as

$$\begin{cases} y = 0, \omega = 0, \frac{\partial^2 \omega}{\partial y^2} = 0 \\ x = 0, x = a, \omega = 0, \frac{\partial \omega}{\partial x} = 0 \\ y = b, \omega = 0, \frac{\partial \omega}{\partial y} = 0 \end{cases} \tag{18}$$

According to the Galerkin method and the boundary conditions of model C, the deflection equation can be given by

$$\omega_3 = \sum_{m=1}^{\infty} \sum_{n=1}^{\infty} A_3 \left(1 - \cos\frac{2m\pi x}{a}\right) \sin^3\frac{n\pi y}{b}, \ (\text{m, n} \in N^+) \tag{19}$$

where $A_3$ is the undetermined constant of model C.

Deriving Equation (12) and substituting it into Equation (19), and then substituting the result into Equation (10), the undetermined constants of model C can be obtained as

$$A_3 = \frac{32a^3b^3}{\pi^4 Dm^3n^3} \cdot \frac{\int_0^a \int_0^b p(x,y)\left(1 - \cos\frac{2m\pi x}{a}\right)\sin^3\frac{n\pi y}{b}dxdy}{135(a/m)^4 + 72(a/m)^2(b/n)^2 + 80(b/n)^4} \tag{20}$$

### 4.3.5. Disturbance Equation of Model D

Model D is shown in Figure 4d using the SCCS. The *LM* and *MN* are clamped by substance coal and rock, and the *OL* and *ON* are simple support. The boundary conditions of model A can be expressed as

$$\begin{cases} x = 0, \omega = 0, \frac{\partial^2 \omega}{\partial x^2} = 0; y = 0, \omega = 0, \frac{\partial^2 \omega}{\partial y^2} = 0 \\ x = a, \omega = 0, \frac{\partial \omega}{\partial x} = 0; y = b, \omega = 0, \frac{\partial \omega}{\partial y} = 0 \end{cases} \tag{21}$$

According to the Galerkin method and the boundary conditions of model D, the deflection equation can be given by

$$\omega_4 = \sum_{m=1}^{\infty} \sum_{n=1}^{\infty} A_4 \sin^3\frac{m\pi x}{a} \sin^3\frac{n\pi y}{b}, \ (\text{m, n} \in N^+) \tag{22}$$

where $A_4$ is the undetermined constants of model D.

Deriving Equation (12) and substituting it into Equation (22), and then substituting the result into Equation (10), the undetermined constants of model D can be obtained as

$$A_4 = \frac{256a^3b^3}{9\pi^4Dm^3n^3} \cdot \frac{\int_0^a \int_0^b p(x,y)\sin^3\frac{m\pi x}{a}\sin^3\frac{n\pi y}{b}dxdy}{25(a/m)^4 + 18(a/m)^2(b/n)^2 + 25(b/n)^4} \tag{23}$$

### 4.4. The Roof Fracture Criterion

When the elastic plate is deflected in the normal direction, the plate will be bent. In this case, the surface of the plate can be simplified into a space curved surface. Because the edge of the sheet is fixed, the arc length in each direction of the plate is greater than the length of the two fixed edges. Because the length of the plate was increased, the stress state of the plate will be mainly tensile stress. Generally, a rock has a high compressive strength and a low tensile strength, and rock shear strength marks the middle of compressive strength and tensile strength. Therefore, the theory of maximal tension stress was used to determine the roof strata fracture.

$$\sigma_u = \sigma_1 \tag{24}$$

where $\sigma_u$ is the ultimate tensile strength of rock, Pa; $\sigma_1$ is the maximum tension stress of the rock, Pa.

In this part, only the small deformations that occur in engineering structures will be considered. The small displacements of particles of a thin plate will usually be resolved into components $dx$ and $dy$ parallel to the coordinate axes $x$ and $y$. These components are assumed to be very small quantities varying continuously over the volume of the body. Consider a small element $dx$ and $dy$ of a thin elastic plate (Figure 5). If the thin plate undergoes a deformation and $dx$ and $dy$ are the components of the displacement of the point 0, the displacement in the $x$-direction of adjacent point A on $x$-axis is $ds_x$. In the same manner, it can be shown that the displacement in the $x$-direction of adjacent point B on $x$-axis is $ds_y$. According to the principle of calculating the arc length, the $ds_x$ and $ds_y$ can be expressed as:

$$\begin{cases} ds_x = \sqrt{1+w'^2}dx = \sqrt{1+\left(\frac{\partial w}{\partial x}\right)^2}dx \\ ds_y = \sqrt{1+w'^2}dy = \sqrt{1+\left(\frac{\partial w}{\partial y}\right)^2}dy \end{cases} \tag{25}$$

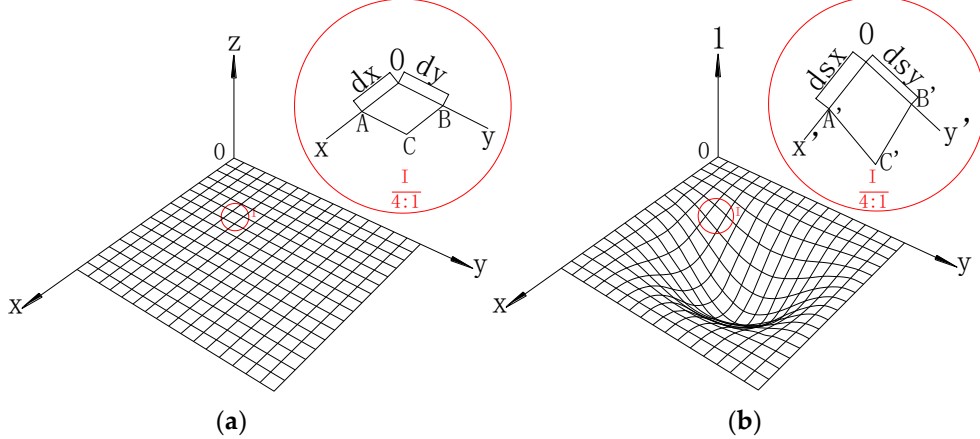

(**a**)　　　　　　　　　　　　　　　　　(**b**)

**Figure 5.** The deformation of the elastic thin plate, (**a**) without deformation; (**b**) small deformation.

The authors shall use the letter $\varepsilon$ for unit elongation. To indicate the directions of strain, the same subscripts to these letters will be used as those for the stress components. The unit elongation of the $x$-direction and the $y$-direction can be expressed as

$$\begin{cases} \varepsilon_x = \frac{ds_x - dx}{dx} = \sqrt{1 + \left(\frac{\partial w}{\partial x}\right)^2} - 1 \\ \varepsilon_y = \frac{ds_y - dy}{dy} = \sqrt{1 + \left(\frac{\partial w}{\partial y}\right)^2} - 1 \end{cases} \tag{26}$$

Based on the generalized Hooke's law and the theory of elastic thin plate, the normal strain of the thin plate can be ignored, i.e., the values of $\varepsilon_z$, $\tau_{xy}$ and $\tau_{yz}$ are zero; thus, the stresses of the $x$-direction and the $y$-direction can be expressed as

$$\begin{cases} \sigma_x = \frac{E}{1-v^2}\left(\varepsilon_x + v\varepsilon_y\right) \\ \sigma_y = \frac{E}{1-v^2}\left(\varepsilon_y + v\varepsilon_x\right) \\ \tau_{xy} = \frac{E}{2(1+v)}\gamma_{xy} \end{cases} \tag{27}$$

Substituting Equation (26) into Equation (27), the relationship between the deflection of the thin plate and the stress at the directions of the $x$-axis and the $y$-axis can be expressed as

$$\begin{cases} \sigma_x = \frac{E}{1-v^2}\left[\left(\sqrt{1 + \left(\frac{\partial w}{\partial x}\right)^2} - 1\right) + v\left(\sqrt{1 + \left(\frac{\partial w}{\partial y}\right)^2} - 1\right)\right] \\ \sigma_y = \frac{E}{1-v^2}\left[\left(\sqrt{1 + \left(\frac{\partial w}{\partial y}\right)^2} - 1\right) + v\left(\sqrt{1 + \left(\frac{\partial w}{\partial x}\right)^2} - 1\right)\right] \\ \tau_{xy} = \frac{E}{2(1+v)}\gamma_{xy} \end{cases} \tag{28}$$

Based on the theory of maximal tension stress, substituting the deflection equation into Equation (28), and solving for the values of $\sigma_x$ and $\sigma_y$, the immediate roof fracture criterion can be expressed as the following:

If the max $[\sigma_x, \sigma_y] < \sigma_u$, then the immediate roof is intact.

If the max $[\sigma_x, \sigma_y] = \sigma_u$, then the immediate roof is the critical level of the fracture and is intact.

If the max $[\sigma_x, \sigma_y] > \sigma_u$, then the immediate roof is fractured. The value of letter '*a*' or letter '*b*' is the fracture span of immediate roof. The high of sub-level is a constant, therefore the value of letter '*a*' can be seen as the first or periodic fracture span of immediate roof.

## 5. The Stress Situation of the Roof through Numerical Simulation Analysis

### 5.1. Numerical Simulation Model

The boundary conditions and stress conditions were both required when the four mechanics model defection equation was calculated. However, because the stress characteristic of immediate roof was difficult to obtain from field measurement tests, FLAC$^{3D}$ was used to calculate the normal stress in the immediate roof. To simulate the normal stress distribution of the four proposed models, the hybrid boundary conditions are applied at the boundary of the model. To study the normal stress distribution of the immediate roof in HGTC, the models are created to simulate the excavation process of HGTC based on the geological condition of the Adaohai Coal Mine.

The immediate roof consists of two or more strata with different lithological characters. The stratums that have the similar mechanical properties are considered as one layer in the simulation model. The Mohr−Coulomb criterion was used to determine the failure of materials in the numerical simulation. Based on the lithological characteristics of Adaohai Coal Mine, the model was simplified to 13 coal or rock layers, as listed in Table 2.

**Table 2.** Primary mechanical parameters of the coal or rock stratum.

| NO. | Lithology | Thickness/m | Density Kg/m³ | Bulk Modulus/GPa | Shear Modulus/GPa | Friction Angle/ Degree (°) | Cohesion/ MPa | Tensile Strength/GPa |
|-----|-----------|-------------|---------------|------------------|-------------------|----------------------------|---------------|----------------------|
| 1 | Loose layer | 20 | 2100 | 7.0 | 3.5 | 25° | 5.5 | 1.6 |
| 2 | Sandy mudstone | Variable | 2600 | 8.1 | 6.0 | 36° | 18.8 | 3.5 |
| 3 | Mudstone | 48 | 2470 | 2.6 | 2.0 | 38° | 4.5 | 1.0 |
| 4 | Sandy mudstone | 24 | 2450 | 8.1 | 6.0 | 36° | 18.8 | 3.5 |
| 5 | Medium sandstone | 16 | 2430 | 10.9 | 6.9 | 31° | 39.5 | 5.1 |
| 6 | Sandstone | 8 | 2600 | 4.9 | 3.7 | 30° | 27.2 | 6.1 |
| 7 | Mudstone | 2 | 2430 | 2.6 | 2.0 | 38° | 4.5 | 1.0 |
| 8 | Coal seam | 28 | 1330 | 1.2 | 0.8 | 28° | 4.2 | 0.9 |
| 9 | Mudstone | 4 | 2400 | 2.6 | 2.0 | 38° | 4.5 | 1.0 |
| 10 | Sandstone | 12 | 2450 | 4.9 | 3.7 | 30° | 27.2 | 6.1 |
| 11 | Medium sandstone | 16 | 2650 | 10.9 | 6.9 | 31° | 39.5 | 5.1 |
| 12 | Sandy mudstone | 24 | 2500 | 8.1 | 6.0 | 36° | 18.8 | 3.5 |
| 13 | Coarse sandstone | 48 | 2500 | 12.5 | 9.4 | 35° | 35.6 | 3.5 |

The normal stress distribution of the immediate roof is affected by many factors. Many experiments are required to analyse these many factors. To simplify the simulation workload, the normal stress distribution of the immediate roof was analysed by the method of orthogonal experiment design [31–33]. Three levels with different depths, coal seam angles, lateral pressure coefficients, and maximum principal stress directions were considered in the orthogonal experiment design to better illustrate the stress distribution of immediate roof in HGTC, as listed in Table 3.

**Table 3.** The simulations of nine representative combinations, based on the orthogonal array L9 (34).

| NO | Seam Depth | Coal Seams Angle | Lateral Pressure Coefficient | Maximum Principal Stress Direction | Length | Width | Height | Blocks Number | Grid Points Number |
|----|-----------|------------------|------------------------------|-----------------------------------|--------|-------|--------|---------------|--------------------|
| 1 | 100 m | 65° | 1 | 0° | 277 m | 200 m | 120 m | 553,750 | 586,921 |
| 2 | 100 m | 75° | 1.25 | 45° | 257 m | 200 m | 120 m | 516,250 | 550,685 |
| 3 | 100 m | 85° | 1.5 | 90° | 239 m | 200 m | 120 m | 515,000 | 549,155 |
| 4 | 300 m | 65° | 1.5 | 45° | 277 m | 200 m | 120 m | 553,750 | 586,921 |
| 5 | 300 m | 75° | 1 | 90° | 257 m | 200 m | 120 m | 516,250 | 550,685 |
| 6 | 300 m | 85° | 1.25 | 0° | 239 m | 200 m | 120 m | 515,000 | 549,155 |
| 7 | 500 m | 65° | 1.25 | 90° | 277 m | 200 m | 120 m | 553,750 | 586,921 |
| 8 | 500 m | 75° | 1.5 | 0° | 257 m | 200 m | 120 m | 516,250 | 550,685 |
| 9 | 500 m | 85° | 1 | 45° | 239 m | 200 m | 120 m | 515,000 | 549,155 |

The variety of mining depths, lateral pressure coefficients and the maximum horizontal principal stress were simulated by applying the different boundary conditions to the simulation model.

A positive correlation was found between the vertical stress and buried depth [34], and different pressures were applied on top of model to simulate various mining depths. The pressure was calculated using equation $\sigma_H = \gamma H$, where $\sigma_H$ is the pressure loading on the top boundary of the model, $H$ is the buried depth, and $\gamma$ is the bulk density of stratum (average value is $2.5 \times 10^4 \, \text{N/m}^3$).

Based on the Mohr−Coulomb criterion and the mechanics of materials [35], the maximum principal stress and its orientation can be calculated as

$$\begin{cases} \sigma_1 = \frac{1}{2}\left(\sigma_x + \sigma_y\right) + \frac{1}{2}\sqrt{\left(\sigma_x - \sigma_y\right)^2 + 4\tau_{xy}{}^2} \\ \sigma_2 = \frac{1}{2}\left(\sigma_x + \sigma_y\right) - \frac{1}{2}\sqrt{\left(\sigma_x - \sigma_y\right)^2 + 4\tau_{xy}{}^2} \\ \alpha_0 = \frac{1}{2}\arctan\left(\frac{-2\tau_{xy}}{\sigma_x - \sigma_y}\right) \end{cases} \tag{29}$$

where $\sigma_1$ and $\sigma_2$ are the maximum principal stress and minimum principal stress in the plane, Pa; $\sigma_x$ and $\sigma_y$ are the stress loading along the *x*-axis and *y*-axis, Pa; $\tau_{xy}$ is the shear stress in the plane, Pa; $\alpha_0$ is the orientation of the maximum horizontal principal stress, degree.

The different maximum principal stress and its orientation angle were simulated using the different boundary conditions. The relationship between the maximum principal stress and the boundary stresses of the simulation model can be expressed as

$$\begin{cases} \sigma_x = \sigma_1 \cos^2 \alpha_0 + \sigma_2 \sin^2 \alpha_0 \\ \sigma_y = \sigma_1 \sin^2 \alpha_0 + \sigma_2 \cos^2 \alpha_0 \end{cases} \tag{30}$$

where $\sigma_x$ and $\sigma_y$ are the stress loading on the left and right boundary of model, Pa.

To determine the relationship between the "$\sigma_1$" and "$\sigma_2$", the statistics of six mines, a total of 12 positions, and the in situ stress parameters were counted. According to the field in situ stress test, the ratio of "$\sigma_1$" to "$\sigma_2$" was 1.74 to 1.96, with an average of 1.89, as listed in Table 4; thus, the ratio of "$\sigma_1$" and "$\sigma_2$" in the orthogonal experiment design is 1.89.

**Table 4.** The in situ stress of the field measurement case in some mines in China.

| In-Situ Stress Test Position | Depth/m | $\sigma_1$/MPa | $\sigma_2$/MPa | $\sigma_H$/MPa | $\sigma_1/\sigma_2$ |
|---|---|---|---|---|---|
| The rock cross-cut at mining level + 1126 at Adaohai Mine | 367.5 | 18.03 | 10.9 | 9.25 | 1.95 |
| The winch chamber at mining level + 1228 at Adaohai Mine | 206.9 | 12.26 | 6.84 | 6.27 | 1.96 |
| The No. 2-1022 tail entry at Ganhe Mine | 461 | 16.18 | 8.38 | 11 | 1.93 |
| The head entry of scope 2 at Ganhe Mine | 529 | 14.78 | 7.92 | 12.77 | 1.87 |
| The No. 2151 head entry at Tuanbai Mine | 33 | 9.27 | 5.32 | 7.8 | 1.74 |
| The No. 310 tail entry at Tuanbai Mine | 405 | 12.37 | 6.7 | 9.67 | 1.80 |
| The No. 10-1021 head entry at Huipodi Mine | 367.9 | 9.32 | 4.99 | 8.77 | 1.87 |
| The haulage roadway of east scope at Huipodi Mine | 355.1 | 10.32 | 5.57 | 8.33 | 1.85 |
| The central of 1051 lane yard at Pangpangta Mine | 490 | 9.63 | 5.1 | 12.26 | 1.89 |
| The No. 1092 tail entry at Pangpangta Mine | 592 | 11.68 | 6.16 | 14.8 | 1.90 |
| The main haulage roadway at Changping Mine | 348.1 | 10.81 | 5.6 | 8.7 | 1.93 |
| The first contact alley at Changping Mine | 343.9 | 9.64 | 4.99 | 8.6 | 1.93 |

In this paper, the stress transferred method was used to simulate the real in situ stress conditions. The boundary conditions for different models are shown in Figure 6.

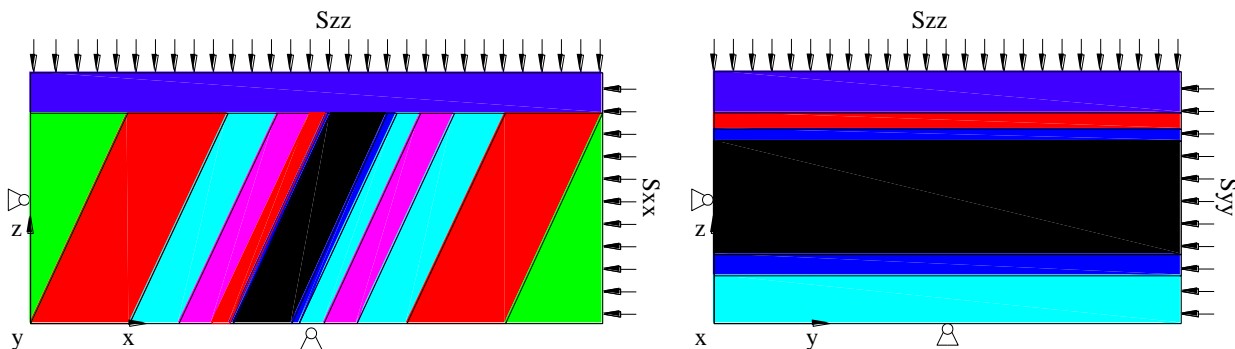

**Figure 6.** The schematic diagram of the boundary condition.

The detailed boundary conditions of the numerical simulation model are shown in Table 5.

**Table 5.** The boundary conditions of the numerical model.

| NO. | The Bottom of Models | The Upper of Models (Mpa) | The Negative of $x$-Axis | The Direction of $x$-Axis (Mpa) | The Negative of $y$-Axis | The Direction of $y$-Axis (Mpa) |
|---|---|---|---|---|---|---|
| 1 | Fixed | 0.25 | Fixed | 0.32 | Fixed | 0.18 |
| 2 | Fixed | 0.25 | Fixed | 0.31 | Fixed | 0.31 |
| 3 | Fixed | 0.25 | Fixed | 0.27 | Fixed | 0.48 |
| 4 | Fixed | 5.25 | Fixed | 7.88 | Fixed | 7.88 |
| 5 | Fixed | 5.25 | Fixed | 6.78 | Fixed | 3.72 |
| 6 | Fixed | 5.25 | Fixed | 8.47 | Fixed | 4.65 |
| 7 | Fixed | 10.25 | Fixed | 9.09 | Fixed | 16.54 |
| 8 | Fixed | 10.25 | Fixed | 19.85 | Fixed | 10.90 |
| 9 | Fixed | 10.25 | Fixed | 10.25 | Fixed | 10.25 |

The distance between the excavation zone and the boundary is 60 m to eliminate the influence of boundaries. The working face was excavated for 4 m each time, for a total of 80 m. The first subsection and second subsection were excavated in turn.

*5.2. Coordinate Conversion*

The stress data calculated by FLAC$^{3D}$ could not be directly used in the proposed thin plate mechanics models because these two methods are based on different coordinate systems. Data conversion was required between these two coordinate systems. The stress information includes the vertical stress ($\sigma_z$), the horizontal stress ($\sigma_x$, $\sigma_y$), and the shear stress ($\tau_{xy}$, $\tau_{xz}$, $\tau_{yz}$). According to the stress status at a point in different Cartesian coordinate system conversion relationships [36], the authors converted the stress data of FLAC$^{3D}$ to data in the new Cartesian coordinate system, for which the $x$-axis is the working face mining distance, the $y$-axis is the inclined direction of immediate roof, and the $z$-axis is the normal direction of immediate roof. The new Cartesian coordinate system was obtained by rotating the numerical simulation Cartesian coordinate system by $\theta$ degrees along the $y$-axis, as shown in Figure 7.

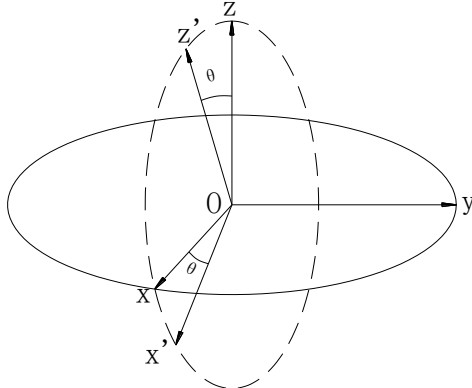

**Figure 7.** The diagram of the SCCS rotation.

The relationship between the different Cartesian coordinate systems can be expressed as

$$\begin{cases} \sigma_{x\prime} = \sigma_x \cos^2 \theta + \sigma_z \sin^2 \theta - \tau_{xz} \sin 2\theta \\ \sigma_{z\prime} = \sigma_x \sin^2 \theta + \sigma_z \cos^2 \theta + \tau_{xz} \sin 2\theta \\ \tau_{x\prime y\prime} = -\tau_{yz} \sin \theta + \tau_{xy} \cos \theta \\ \tau_{y\prime z\prime} = \tau_{yz} \cos \theta + \tau_{xy} \sin \theta \\ \tau_{x\prime z\prime} = (\sigma_x - \sigma_z) \sin \theta \cos \theta + \tau_{xz} \cos 2\theta \end{cases} \quad (31)$$

where $\sigma_{x\prime}$, $\sigma_{z\prime}$, $\tau_{x\prime y\prime}$, $\tau_{y\prime z\prime}$, and $\tau_{x\prime z\prime}$ are stress and strain in the new Cartesian coordinate system.

### 5.3. The Normal Stress Distribution of the Immediate Roof

Normal stress distribution, in which the *z*-axis is the normal stress direction, is given by Equation (31). The normal stress of the immediate roof along the direction of dip with different excavation conditions is shown in Figure 8.

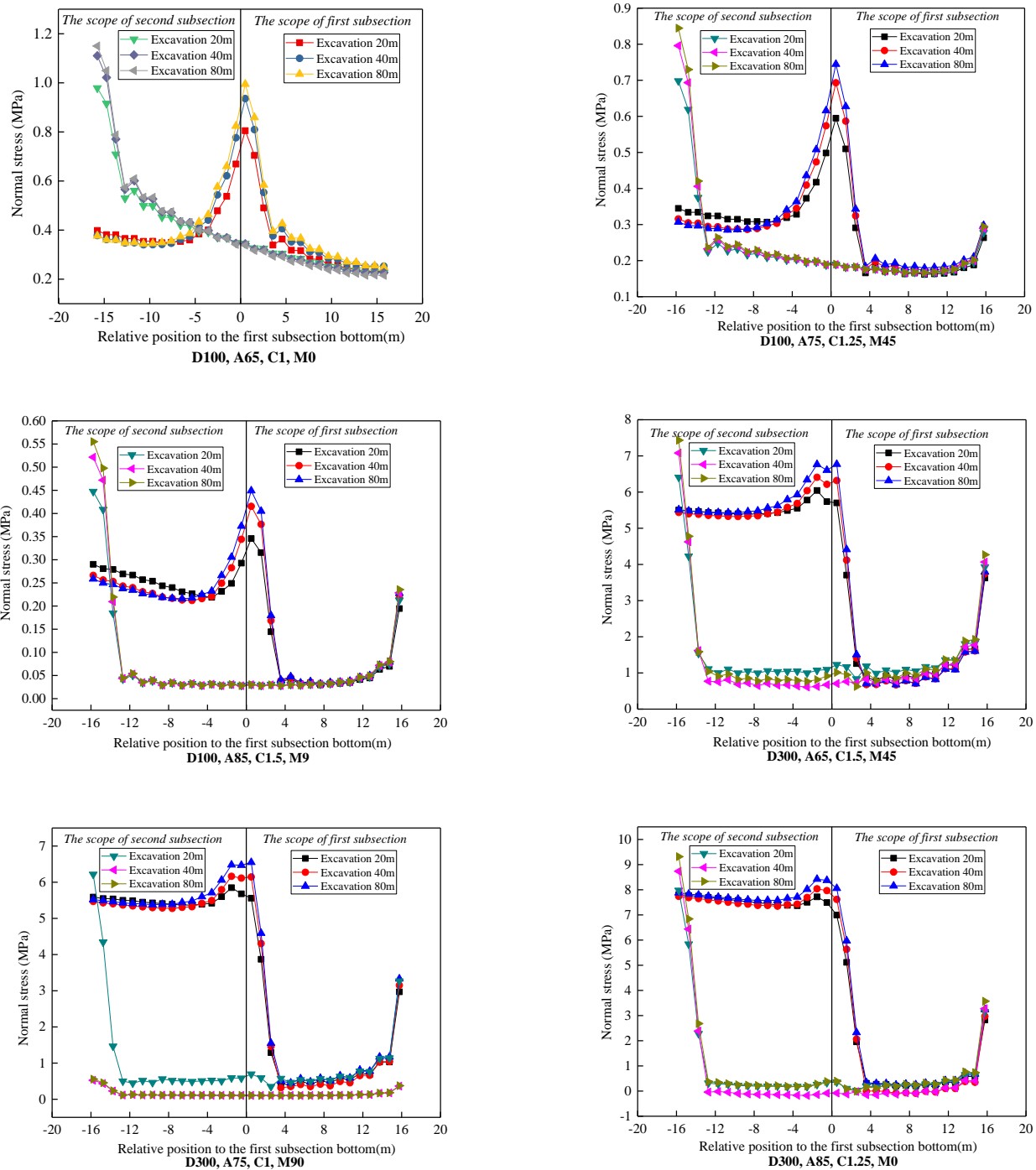

**Figure 8.** *Cont*.

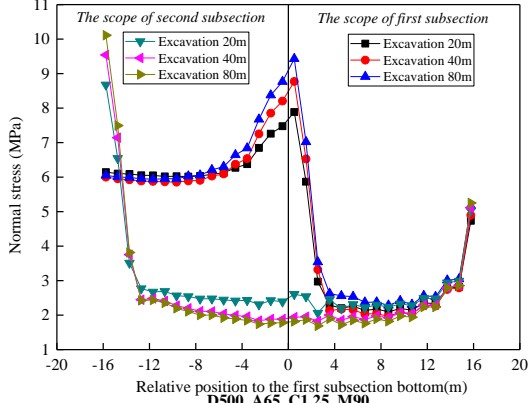

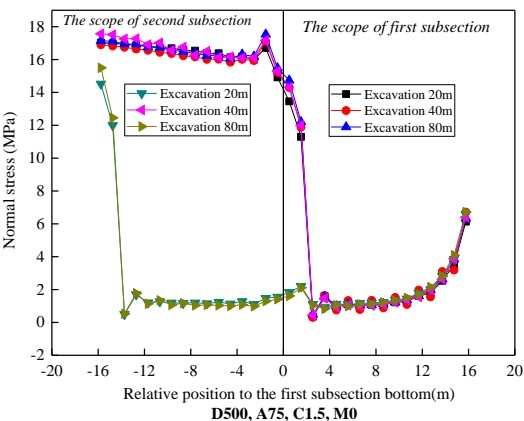

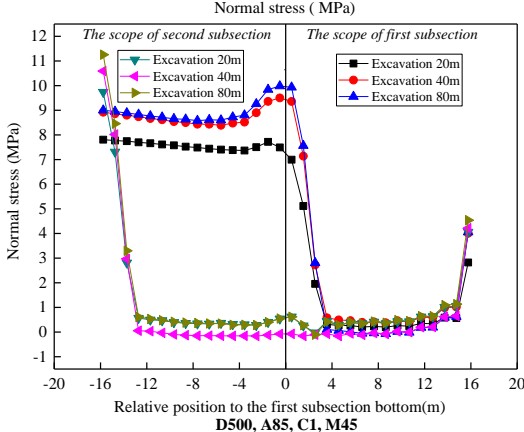

**Figure 8.** The normal stress distribution of the immediate roof in the direction of dip (D is the seam depth, A is the coal seam angle, C is the lateral pressure coefficient, and M is the maximum principal stress direction).

Figure 8 shows the normal stress distribution of the immediate roof when the working face was excavated 20 m, 40 m, and 80 m in the first subsection and the second subsection. The following normal stress regularity of the immediate roof can be drawn: the normal stress distribution of the first and second subsection immediate roof can be approximately considered as a uniform distribution; the normal stress is positively associated with the seam depth, the lateral pressure coefficient, and the maximum principal stress direction (the angle is between 0 and 90), whereas it is negatively associated with the coal seam angle. The influences in descending order of the four orthogonal factors on the simulation results in a linear distribution scope of immediate roof is as follows: the coal seam angle, the seam depth, the maximum principal stress direction, and the lateral pressure coefficient. The influence in descending order of the four orthogonal factors on the simulation results of the uniform distribution scope of immediate roof is as follows: the maximum principal stress direction, the coal seam angle, the seam depth, and the lateral pressure coefficient.

Considering the effect of four factors including the depths, coal seam angles, lateral pressure coefficients, and maximum principal stress directions on the normal stress distribution of immediate roof, although the above four factors have a significant difference in the normal stress, the distribution of normal stress is similar and can be regarded as uniform load.

*5.4. The Fractures Span of the Immediate Roof in HGTC*

Through the above study, the deflection equation of four thin models, the normal stress distribution of the immediate roof, and the failure criterion were obtained. Based on these results, the fractures span of the immediate roof in HGTC can be obtained.

In this part, the fractures span of model A is analysed and demonstrated. According to the results of the numerical simulation, the normal stress distribution can be simplified as uniform loading: $p(x,y) = p$. In addition, the deflection equation of model A under the uniform load can be expressed as

$$\omega_1 = A_1 \left(1 - \cos\frac{2\pi x}{a}\right)\left(1 - \cos\frac{2\pi y}{b}\right) = \frac{a^4 b^4 p}{4\pi^4 D(3a^4 + 2a^2 b^2 + 3b^4)}\left(1 - \cos\frac{2\pi x}{a}\right)\left(1 - \cos\frac{2\pi y}{b}\right) \quad (32)$$

Substituting Equation (32) into Equation (28), the tensile stress along the *x*-axis (the strike of immediate roof) and the *y*-axis (the dip of immediate roof) can be expressed as

$$\begin{cases} \sigma_{tx1} = \frac{E}{1-v^2}\left[\left(\sqrt{1 + \frac{16A_1{}^2\pi^2}{a^2}\sin^4\frac{\pi y}{b}\sin^2\frac{2\pi x}{a}} - 1\right) + v\left(\sqrt{1 + \frac{16A_1{}^2\pi^2}{b^2}\sin^4\frac{\pi x}{a}\sin^2\frac{2\pi y}{b}} - 1\right)\right] \\ \sigma_{ty1} = \frac{E}{1-v^2}\left[\left(\sqrt{1 + \frac{16A_1{}^2\pi^2}{b^2}\sin^4\frac{\pi x}{a}\sin^2\frac{2\pi y}{b}} - 1\right) + v\left(\sqrt{1 + \frac{16A_1{}^2\pi^2}{a^2}\sin^4\frac{\pi y}{b}\sin^2\frac{2\pi x}{a}} - 1\right)\right] \end{cases} \quad (33)$$

where $\sigma_{tx1}$ and $\sigma_{ty1}$ are the tensile stress of the *x*-axis and *y*-axis, Pa.

The maximum value of Equation (33) was calculated, where the corresponding point $(x, y)$ denotes the position of the maximum tensile stress in the immediate roof before first fracture.

When the value of "*x*" was "$0.3524a$" or "$0.6476a$" and the value of "*y*" was "$\frac{b}{\pi}\arctan\frac{2va}{b}$" or "$b - \frac{b}{\pi}\arctan\frac{2va}{b}$", the maximum tensile stress ($\sigma_{tx1max}$) in the direction of the *x*-axis can be obtained:

$$\sigma_{tx1max} = \frac{E}{1-v}\left(\sqrt{1 + \frac{256A^2\pi^2 v^4 a^2}{(4v^2 a^2 + b^2)^2}} - 1\right) \quad (34)$$

where the value of "*x*" was "$\frac{a}{\pi}\arctan\frac{2vb}{a}$" or "$a - \frac{a}{\pi}\arctan\frac{2vb}{a}$", m; the value of "*y*" was "$0.3524b$" or "$0.6476b$",m; the maximum tensile stress ($\sigma_{ty max}$) in the direction of the *y*-axis can be obtained by

$$\sigma_{ty1max} = \frac{E}{1-v}\left(\sqrt{1 + \frac{256A^2\pi^2 v^4 b^2}{(4v^2 b^2 + a^2)^2}} - 1\right) \quad (35)$$

The symbol b represents the inclined extent of the first subsection and is a constant. When $\sigma_{tx max}$ or $\sigma_{ty max}$ reaches the ultimate tensile strength of immediate roof rock, the immediate roof is fractured. The symbol a is the first fracture span of the immediate roof in the first subsection.

Similarly, the periodic fractures span of the first subsection and the first and periodic fractures span of the second subsection can be obtained.

For Model B, when the value of "*x*" was "$\frac{a}{\pi}\arctan\frac{3b}{2av}$", m; the value of "*y*" was "$0.25b$" or "$0.75b$", m; the maximum tensile stress ($\sigma_{tx2max}$) in the direction of *x*-axis can be obtained by

$$\sigma_{tx2max} = \frac{E}{(1-v^2)}\left[\left(\sqrt{1 + \frac{2916A_2{}^2 b^4 a^2 v^2}{(9b^2 + 4a^2 v^2)^3}} - 1\right) + v\left(\sqrt{1 + \frac{2916A_2{}^2 b^6}{(9b^2 + 4a^2 v^2)^3}} - 1\right)\right] \quad (36)$$

In addition, when the value of "$x$" was "$\frac{a}{\pi}\arctan\frac{3bv}{2a}$, m; the value of "$y$" was "0.25$b$" or "0.75$b$", m; the maximum tensile stress ($\sigma_{ty2\max}$) in the direction of the $y$-axis can be obtained by

$$\sigma_{ty2\max} = \frac{E}{1-v^2}\left[\left(\sqrt{1+\frac{2916A_2{}^2b^6v^6}{(9b^2v^2+4a^2)^3}}-1\right)+v\left(\sqrt{1+\frac{2916A_2{}^2b^4a^2v^2}{(9b^2v^2+4a^2)^3}}-1\right)\right] \quad (37)$$

For Model C, when the value of "$x$" was "0.25$a$" or "0.75$a$, m; the value of "$y$" was "$\frac{b}{\pi}\arctan\frac{3av}{2b}$", m; the maximum tensile stress ($\sigma_{tx3\max}$) in the direction of the $x$-axis can be obtained by

$$\sigma_{tx3\max} = \frac{E}{1-v^2}\left[\left(\sqrt{1+\frac{2916A_2{}^2a^6v^6}{(9a^2v^2+4b^2)^3}}-1\right)+v\left(\sqrt{1+\frac{2916A_2{}^2a^4b^2v^2}{(9a^2v^2+4b^2)^3}}-1\right)\right] \quad (38)$$

In addition, when the value of "$x$" was "0.25$a$" or "0.75$a$", m; the value of "$y$" was "$\frac{b}{\pi}\arctan\frac{3a}{2bv}$", m; the maximum tensile stress ($\sigma_{ty3\max}$) in the direction of the $y$-axis can be obtained by

$$\sigma_{ty3\max} = \frac{E}{(1-v^2)}\left[\left(\sqrt{1+\frac{2916A_2{}^2a^4b^2v^2}{(9a^2+4b^2v^2)^3}}-1\right)+v\left(\sqrt{1+\frac{2916A_2{}^2a^6}{(9a^2+4b^2v^2)^3}}-1\right)\right] \quad (39)$$

For Model D, when the value of "$x$" was "0.9$a$", m; the value of "$y$" was "0.9$b$", m; the maximum tensile stress ($\sigma_{tx4\max}$ and $\sigma_{ty4\max}$) in the directions of the $x$-axis and the $y$-axis can be obtained by

$$\sigma_{tx4\max} = \frac{E}{1-v^2}\left\{\left[\sqrt{1+\frac{9A_4{}^2m^2\pi^2(9-3\sqrt{5})}{4a^2}}-1\right]+v\left(\sqrt{1+\frac{9A_4{}^2n^2\pi^2(9-3\sqrt{5})}{4b^2}}-1\right)\right\} \quad (40)$$

$$\sigma_{ty4\max} = \frac{E}{1-v^2}\left\{\left(\sqrt{1+\frac{9A_4{}^2n^2\pi^2(9-3\sqrt{5})}{4b^2}}-1\right)+v\left[\sqrt{1+\frac{9A_4{}^2m^2\pi^2(9-3\sqrt{5})}{4a^2}}-1\right]\right\} \quad (41)$$

In the actual engineering application process, the Poisson's ratio "$v$" of the immediate roof, the rock layer inclination "$\theta$", the segmental height "$b\sin\theta$", the elastic modulus "$E$," and the normal stress of the immediate roof can be regarded as constants, so the difference between the advancing distance and the working face can be obtained. The maximum tensile stress "$\sigma_{tx\max}$" and "$\sigma_{ty\max}$" pull in different directions on the immediate roof. At the same time, the tensile strength "$\sigma_u$" and strength reduction coefficient "$\beta$" of the immediate roof can also be obtained through laboratory and field sonic tests. The maximum tensile stress of the immediate roof obtained by theoretical calculation and the maximum tensile stress of the rock obtained by the test are compared and analysed. According to the rock formation failure criterion established above, the stability of the immediate roof is judged. When the internal tensile stress of the immediate roof reaches the tensile strength of the rock mass, the tensile failure of the immediate roof will occur at the maximum tensile stress, and the internal cracks of the immediate roof will expand. The direction is perpendicular to the direction of tensile stress, and the corresponding advancing distance of the working face is the first or periodic failure span of the immediate roof rock stratum.

## 6. Discussion

From the perspective of rock depositional age, the rock layer above the coal seam and formed after the formation of the coal seam is generally called the roof. From the perspective of engineering practice, the rock layer above the coal seam is generally called the roof. The roof of the near-upright extra-thick coal seam is composed of three types of

coal-rock masses: the "roof side" rock layer, the "overlying coal seam," and the "floor side" rock layer. The roof is no longer a single rock layer roof in the traditional sense. This is the essential difference between the near-vertical and extra-thick coal seam roof and the traditional gentle and inclined coal seam roof [37]. In the process of mining near-vertical and extra-thick coal seams, not only will the "roof side" rock layers and "overlying coal seams" move and collapse, but also the "bottom side" rock layers that are more complicated than inclined and gently inclined coal seams. There is a problem of slippage or overturning failure of the roof. With the increase of the inclination of the coal seam, the influence of the damage of the direct roof on the stope and the roadway is gradually highlighted, and the influence of the damage of the basic roof on the stope and the roadway is gradually weakened. The damage of the direct roof of the near-upright coal seam will cause a large area of coal and rock mass to suddenly rush to the working face and two roadways, resulting in rapid deformation of the roadway, overturning of equipment, overturning of personnel, and even serious ground pressure disaster accidents, which will threaten the safety of coal mines. Efficient production poses a serious threat. Therefore, it is of great significance to study the breaking position of the first (periodic) breaking of the direct roof and the evolution process of breaking during the mining of near-vertical coal seams.

In this work, the four-edge-clamped plate model and the top-three-clamped-edges simply supported plate model are introduced to calculate the deflection of the immediate roof before the immediate roof first fracture of the first subsection or the other subsections. The bottom-three-clamped-edges simply supported plate model and the two-adjacent-edges clamped on the edge of a simply supported plate model are introduced to calculate the deflection of the immediate roof before the immediate roof periodic fracture of the first subsection or the other subsections. The immediate roof is assumed to be an elastic thin plate before the first and periodic fracture, and the Galerkin method is used to calculate the solution of deflection equation under the effect of normal stress. By introducing the new fracture criteria, the maximum tensile stress strength criterion and generalized Hooke's law, the fracture processes of the immediate roof of these four models are analysed. To verify the normal stress distribution of the four thin plate models, the FLAC$^{3D}$ numerical simulation software was used to simulate the direct top stress distribution characteristics of the horizontal segmented top coal caving in the near-upright extra-thick coal seam. According to the simulation results, the normal stress distribution of the immediate roof is uniform. Thus, the deflection equations can be calculated under the uniform normal stress. The most dangerous position of the immediate roof and the first or periodic fracture span can be obtained by using the thin plate model and the new fracture criteria. It can better predict the weighting situation of the working face, and ensure the safe, efficient, and sustainable mining of coal mines.

This paper has academic and practical significance for further research on the roof stability of the near-upright extra thick coal seam. However, some deficiencies need to be pointed out. Due to the limitations of experimental conditions, in the process of solving the thin plate mechanical model, only the influence of the stress on the normal line of the thin plate on the deflection of the plate is considered. The influence of the tangential stress on the deflection of the thin plate is ignored. The result has certain limitations. The number of coal mining faces that are near-vertical and have an extra-thick coal seam using the horizontally grouped top-coal drawing method is relatively small in China. This is not conducive to the development of on-site measurement work, and the theoretical model lacks the verification of on-site engineering practice, which needs more targeted research in the future.

## 7. Conclusions

The objective of this work was to investigate the roof fracture regularity of near-vertical and extremely thick coal seams with HGTC. The Galerkin method, the theory of maximal tension stress, and orthogonal experiment design were used to calculate the roof before the first and periodic fractures span. The following main conclusion can be drawn:

(1) Four mechanical models and the deflection equations for roof fractures in near-vertical and extremely thick coal seams have been established, which cover the possibility of stress on the roof when the roof is gradually damaged by a horizontally grouped top-coal drawing method in near-vertical and extremely thick coal seams.

(2) The influence of the seam depth, the coal seam angle, the lateral pressure coefficient, and the maximum principal stress direction on the normal stress distribution of the immediate roof with HGTC were analysed by orthogonal experiment design and FLAC$^{3D}$, and the normal stress could be simplified as uniform loading. The load could be set as a uniform load according to the numerical simulation results, when calculating the maximum tensile stress of the roof under the four states.

(3) A calculation method for roof stress distribution based on Hooke's law and arc length theorem is proposed. Taking the maximum tensile stress strength criterion as the near-vertical immediate roof fracture criterion, the first fracture and periodic fractures span could be obtained, which can better predict the weighting situation of the working face, and ensure the safe, efficient, and sustainable mining of coal mines.

**Author Contributions:** Conceptualization, G.Z. and Z.X.; methodology, G.Z.; investigation, G.Z. and Q.L.; writing-original draft, G.Z., Q.L. and Z.X.; formal analysis, G.Z., Z.X. and Y.Z.; writing—review & editing, G.Z. and Y.Z. All authors have read and agreed to the published version of the manuscript.

**Funding:** This research was funded by the state Key Research and Development Program of China (2018YFC0604501), the Science and Technology Innovation Project of China Energy Investment Corporation (SHGF-16-24), and the National Natural Science Foundation of China (51904303, 52004291).

**Institutional Review Board Statement:** Not applicable.

**Informed Consent Statement:** Not applicable.

**Data Availability Statement:** The data used to support the findings of this study are available from the corresponding author upon request.

**Conflicts of Interest:** The authors declare no conflict of interest.

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
