# Peer review of "Roof Fractures of Near-Vertical and Extremely Thick Coal Seams in Horizontally Grouped Top-Coal Drawing Method Based on the Theory of a Thin Plate"

_sustainability, doi:10.3390/su141610285_

Round 1
Reviewer 1 Report
Dear Authors,
Congratulations on your work, which is focused on a very interesting subject. As any other paper in this phase, there are some amendments to do, whose can improve the overall quality of your paper. Thus, I'm providing below some comments and suggestions, trying to collaborate by this way in improving your paper:
1. The Abstract doesn't clearly state the literature gap found, as well as the main motivation to develop this work. Thus, please clearly state the gap found in the literature in the Abstract, Introduction and Conclusions. The mains goals are also not clear in the Abstract.
2. The novelty brought by your work is also not properly pointed out. Thus, please state clearly the novelty that your paper represents for the scientific community, stating as well if your contribution is exclusively scientific or if there was some practical motivation behind the development of your work. Any industrial application based on this work should also be pointed out.
3. Regarding the Abstract, it is not possible to correlate the work done with sustainability. Thus, please correlate the main goals of your work with sustainable goals.
4. No clear conclusions are presented in the Abstract that holds the reader to your paper. Please consider to improve this.
5. All keywords are composed and very big. This will make difficult to find your paper. Once again, sustainability seems does not be important for this work.
6. The name of referred Authors are completely in capitals. Why?
7. Please avoid the use of large number of references to a single idea. There are some cases those need to be rearranged.
8. Some acronyms are used in the text for the first time without the corresponding extended name (S-R, HGTC (this is only explained in the keywords, which is not the best place to do it)).
9. In Table 1, please insert a space between values and units.
10. When describing the variables contained in each formula, please point out the units using the International Units System.
11. The Discussion of the results doesn't compare ideas or values now obtained with others previously referred in the literature review. Thus, this discussion is unuseful.
12. No experimental validation of the model developed is presented.
13. Regarding the Conclusions, the novelty is difficult to understand and no correlation to SUSTAINABILITY is found.
Kind regards.
Author Response
Thanks for your valuable suggestion. Those comments are all valuable and very helpful for revising and improving our paper. We have read comments carefully and revised accordingly. Below are the responds to the reviewers’ comments:
Comment 1: The Abstract doesn't clearly state the literature gap found, as well as the main motivation to develop this work. Thus, please clearly state the gap found in the literature in the Abstract, Introduction and Conclusions. The mains goals are also not clear in the Abstract.
Response: Thanks for your valuable suggestion. In this paper, according to the opinions of reviewers, we revised the abstract of the paper to supplement the purpose, significance and objective of this work.
Comment 2: The novelty brought by your work is also not properly pointed out. Thus, please state clearly the novelty that your paper represents for the scientific community, stating as well if your contribution is exclusively scientific or if there was some practical motivation behind the development of your work. Any industrial application based on this work should also be pointed out.
Response: Thanks for your valuable suggestion. In this paper, according to the opinions of reviewers, some related professional literatures have been added in the introduction part. At the same time, it also complements the purpose, significance and objective of this work, as well as the industrial application value in the introduction and abstract.
Comment 3: Regarding the Abstract, it is not possible to correlate the work done with sustainability. Thus, please correlate the main goals of your work with sustainable goals.
Response: Thanks for your valuable suggestion. The abstract has been revised to further highlight the relevance between the main objectives of our work and the sustainability goals.
Comment 4: No clear conclusions are presented in the Abstract that holds the reader to your paper. Please consider to improve this.
Response: Thanks for your valuable suggestion. The Abstract has been revised while supplementing the important conclusions of the work.
Comment 5: All keywords are composed and very big. This will make difficult to find your paper. Once again, sustainability seems does not be important for this work.
Response: Thanks for your valuable suggestion. The keywords have been revised.
Comment 6: The name of referred Authors are completely in capitals. Why?
Response: Thanks for your valuable suggestion. The name of referred Authors have been revised.
Comment 7: Please avoid the use of large number of references to a single idea. There are some cases those need to be rearranged.
Response: Thanks for your valuable suggestion. Revised the citation format of the references and rearranged the citations.
Comment 8: Some acronyms are used in the text for the first time without the corresponding extended name (S-R, HGTC (this is only explained in the keywords, which is not the best place to do it)).
Response: Thanks for your valuable suggestion. All acronyms in the text have been revised.
Comment 9: In Table 1, please insert a space between values and units.
Response: Thanks for your valuable suggestion. Table 1 has been modified to insert a space between the value and the unit
Comment 10: When describing the variables contained in each formula, please point out the units using the International Units System.
Response: Thanks for your valuable suggestion. Added SI units for corresponding variables when describing all formula variables.
Comment 11: The Discussion of the results doesn't compare ideas or values now obtained with others previously referred in the literature review. Thus, this discussion is unuseful.
Response: Thanks for your valuable suggestion. Modified the Discussion section to further clarify the research value of this work
Comment 12: No experimental validation of the model developed is presented.
Response: Thanks for your valuable suggestion. The established model is compared with the actual production situation on site. In this works, the roof fracture regularity of the near-vertical and extremely thick coal seams with Horizontally Grouped Top-Coal drawing method was studied using the theory of an elastic thin plate. Four mechanical models of the immediate roof were established based on the different mining conditions before the roof fractures with the immediate roof assumed to be in a state of elasticity. The orthogonal experiments designs were used to analyse the normal stress distribution of the immediate roof. The immediate roof normal stress distributions of the FLAC3D are substituted in the four thin plate models. By introducing the fracture criteria, the four models can be applied to simulate the fracture process of the immediate roof with Horizontally Grouped Top-Coal drawing method.
Comment 13:. Regarding the Conclusions, the novelty is difficult to understand and no correlation to SUSTAINABILITY is found.
Response: Thanks for your valuable suggestion. The conclusion section has been revised to further clarify the main innovations of the article. The main highlights of this work are: (1) Four mechanical models for roof fractures in suberect and extremely thick coal seams have been established; (2)A calculation method for roof stress distribution based on Hooke’s law and arc length theorem is proposed; (3) Taking the maximum tensile stress strength criterion as the suberect immediate roof fracture criterion, the first fracture and periodic fractures span could be obtained. It has been refined in the light of expert opinions. The research in this paper can promote the continuous safe production of coal mines.
Reviewer 2 Report
Good and valuable research has been done.
In this study, to analyses the fracture process during sub erect and extremely thick coal seam mining, the four-edge-clamped plate model and the top-three-clamped-edges simply supported plate model are introduced to calculate the deflection of the immediate roof before the immediate roof first frac- ture of the first subsection or the other subsections. The bottom-three-clamped-edges simply sup- ported plate model and the two-adjacent-edges clamped on the edge of a simply supported plate model are introduced to calculate the deflection of the immediate roof before the immediate roof periodic fracture of the first subsection or the other subsections. The immediate roof is assumed to be an elastic thin plate before the first and periodic fracture, and the Galerkin method is used to calculate the solution of deflection equation under the effect of normal stress. By introducing the new fracture criteria, the maximum tensile stress strength criterion and generalized Hooke’s law, the fracture processes of the immediate roof of these four models are analysed. To verify the normal stress distribution of the four thin plate models, FLAC3D software is used to perform the following tasks: simulate the mining process in the steeply inclined thick coal seam mining face; undertake anorthogonal numerical simulation experiment in three levels with different depths, coal seam angles, lateral pressure coefficients and orientations of maximum horizontal principal stress; translate the immediate roof stress of the corresponding 9 simulation experiments into suberect immediate roof normal stress; and use the above results to determine the distribution regularity of normal stress along the dip direction of the roof under the circumstance of different advancing distances and different subsections. According to the simulation results, the normal stress distribution of the im- mediate roof can be seen as uniform. Thus, the deflection equations can be calculated under the uniform normal stress, and the most dangerous position of the immediate roof and the first or pe- riodic fracture span can be obtained by using the thin plate model and the new fracture criteria.
The manuscript is well organized and has good content.
- Authors are recommended to emphasis the novelty and significance of the study in more detail.
- Authors are recommended to discuss the obtained results with literature in more detail.
- Some Figures, such as Figure 7, are marked as dashed lines or colors that are clearer in print.
- Please use the appropriate font size in Figure 7. texts are not legible.
- Topic “Disturbance equation of the mechanics models” is of considerable importance. Provide a more complete explanation on page 6.
- Some parts of Figures 7 and 3 is not clear. It is recommended to use a higher quality image and the text should be shown in a more appropriate color in the image.
- In Figures 2 and 3, the writings and lines of the figures intersect. Modify the shapes so that these intersections are removed.
- Authors are recommended to provide a more complete explanation in the discussion and conclusions sections.
- Some of the references provided are old. It is suggested that a number of recent papers that are new and have been published in the last five years be used in the introduction and references list and about fractures.
Author Response
Thanks for your valuable suggestion. Those comments are all valuable and very helpful for revising and improving our paper. We have read comments carefully and revised accordingly. Below are the responds to the reviewers’ comments:
Comment 1: Authors are recommended to emphasis the novelty and significance of the study in more detail.
Response: Thanks for your valuable suggestion. In this paper, according to the opinions of reviewers, the novelty and significance of the study has been added to the abstract and introduction.
Comment 2: Authors are recommended to discuss the obtained results with literature in more detail.
Response: Thanks for your valuable suggestion. In this paper, according to the opinions of reviewers, the relevant research results are discussed in detail in the introduction.
Comment 3: Some Figures, such as Figure 7, are marked as dashed lines or colors that are clearer in print.
Response: Thanks for your valuable suggestion. The line shapes and colors of all figures in the text have been modified.
Comment 4: Please use the appropriate font size in Figure 7. texts are not legible.
Response: Thanks for your valuable suggestion. The text size in Figure 7 has been modified.
Comment 5: Topic “Disturbance equation of the mechanics models” is of considerable importance. Provide a more complete explanation on page 6.
Response: Thanks for your valuable suggestion. A more complete description of " Disturbance equation of the mechanics models " has been added.
Comment 6: Some parts of Figures 7 and 3 is not clear. It is recommended to use a higher quality image and the text should be shown in a more appropriate color in the
Response: Thanks for your valuable suggestion. Figures 7 and 3 have been replaced, and the color and clarity of the pictures have been adjusted.
Comment 7: In Figures 2 and 3, the writings and lines of the figures intersect. Modify the shapes so that these intersections are removed.
Response: Thanks for your valuable suggestion. Figures 2 and 3 have been replaced, and the text and lines in the figures have been modified.
Comment 8: Authors are recommended to provide a more complete explanation in the discussion and conclusions sections.
Response: Thanks for your valuable suggestion. In this paper, according to the opinions of reviewers, A more complete explanation has been provided in the Discussion and Conclusions section.
Comment 9: Some of the references provided are old. It is suggested that a number of recent papers that are new and have been published in the last five years be used in the introduction and references list and about fractures.
Response: Thanks for your valuable suggestion. In this paper, according to the opinions of reviewers, some related professional literatures about fractures in the last five years have been added in the introduction part.
Round 2
Reviewer 1 Report
Dear Authors,
Thank you so much for your effort in properly addressing the Reviewers' comments and suggestions. Now, just two improvements are needed:
1. (FORMAT) Please insert a space between the text and the square brackets of the references.
2. (TECHNICAL/MANDATORY) Please insert a DISCUSSION section where your results are compared to others previously obtained in similar works or attempts to solve similar problems.
Kind regards.
Author Response
Dear reviewer:
We would like to submit the revised manuscript entitled Roof Fractures of Suberect and Extremely Thick Coal Seams in Horizontally Grouped Top-coal Drawing Method Based on the Theory of a Thin Plate” (ID: sustainability-1779261). Those comments are all valuable and very helpful for revising and improving our paper. We have read comments carefully and revised accordingly.In this work, we revised the formats of references cited in the text, checked the relevance of the indexed references in the text, and added a discussion part in the text, focusing on the research background, research objectives, main research ideas and existing deficiencies of the research results obtained in this paper. The paper has been proofread by English speaking person, who is a teacher majoring in English at Datong University in Shanxi Province of China.
We tried our best to improve the manuscript and made some changes in the manuscript. These changes will not influence the content and framework of the paper. And here we did not list the changes but marked in revised paper.
We appreciate for your warm work earnestly, and hope that the correction will meet with approval.
Thank you and best regards.
Yours sincerely,
Guojun Zhang

Reviewer 2 Report
The desired corrections have been made. In my opinion, the article can be accepted.
Author Response
Dear reviewer:
We would like to submit the revised manuscript entitled Roof Fractures of Suberect and Extremely Thick Coal Seams in Horizontally Grouped Top-coal Drawing Method Based on the Theory of a Thin Plate” (ID: sustainability-1779261). Those comments are all valuable and very helpful for revising and improving our paper. We have read comments carefully and revised accordingly.
In this work, we revised the formats of references cited in the text, checked the relevance of the indexed references in the text, and added a discussion part in the text, focusing on the research background, research objectives, main research ideas and existing deficiencies of the research results obtained in this paper. The paper has been proofread by English speaking person, who is a teacher majoring in English at Datong University in Shanxi Province of China.
We tried our best to improve the manuscript and made some changes in the manuscript. These changes will not influence the content and framework of the paper. And here we did not list the changes but marked in revised paper.
We appreciate for your warm work earnestly, and thank you very much for your recognition of our work.
Thank you and best regards.
Yours sincerely,
Guojun Zhang
E-mail: cherish-guojun@hotmail.com
